# Community deployment of metofluthrin emanators to control indoor *Aedes aegypti*: Efficacy results from a crossover trial in Yucatan, Mexico

Azael Che-Mendoza[1], Guillermo Guillermo-May[1], Oscar D. Kirstein[2], Aylin Chi-Ku[1], Norma Pavía-Ruz[3], Anuar Medina-Barreiro[1], Gabriela González-Olvera[1], Gregor Devine[4], Gonzalo Vazquez-Prokopec[2], Pablo Manrique-Saide[1]*

1 Unidad Colaborativa para Bioensayos Entomologicos, Universidad Autonoma de Yucatan, Merida, Mexico, 2 Department of Environmental Sciences, Emory University, Atlanta, Georgia, United States of America, 3 Centro de Investigaciones Regionales Unidad Biomedicas, Universidad Autonoma de Yucatan, Merida, Mexico, 4 Mosquito Control Laboratory, QIMR Berghofer Medical Research Institute, Brisbane, Queensland, Australia

☯ These authors contributed equally to this work.
* pablo_manrique2000@hotmail.com

## Abstract

### Background

Spatial emanators (SE) are innovative tools for controlling indoor *Aedes aegypti* due to their relatively easy use and high efficacy. Large-scale implementation challenges include community adoption, particularly ensuring proper installation and timely replacement as SE efficacy wanes.

### Methodology and principal findings

We conducted a three-arm, open-label entomological cluster randomized controlled trial with a crossover design, involving 588 households, to assess the entomological effect of the community use of metofluthrin emanators. Arms were: "no treatment"; "community-led deployment" (CD), where the households were responsible for installing and replacing SE with minimal guidance; and "managed deployment" (MD), where the research team handled SE installation and replacement. Emanators were replaced every 3 weeks across four deployment cycles, followed by a crossover between the CD and MD arms. Indoor resting mosquitoes were collected using Prokopack aspirators, and human landing counts (HLCs) were conducted in a subset of 12 houses (4 by arm) at the first, fourth, fifth, and eighth SE replacement rounds. Values of each endpoint during all sampling periods were compared using generalized linear mixed effects models (GLMM), the coefficients of the best-fitting model estimated that SE intervention reduced the number of *Ae. aegypti* per house by 32.7% (95%CI = 16.2-46.0%) in the CD arm and 36.8% (21.1-49.3%) in the MD

**Data availability statement:** All data are in the manuscript and/or supporting information files.

**Funding:** This study was supported by a grant from Innovative Vector Control Consortium (D.G, PI). The funders had no role in study design, data collection and analysis, decision to publish, or preparation of the manuscript.

**Competing interests:** The authors have declared that no competing interests exist.

arm. HLCs accounted 74–94% efficacy (MD) and 35–79% (CD). The crossover analysis found no significant difference between periods and arms, demonstrating the community's ability to manage SE as effectively as research team, even without prior training.

## Conclusions/significance

This trial suggests that safe, portable SE are suited to deployment by householders as a rapid response to local *Aedes*-borne disease outbreaks even in the presence of high pyrethroid resistance in the local *Aedes* population. In urban areas where effective coverage and resourcing is a challenge to control campaigns, community "ownership" of SE products may enhance the impact of insecticidal interventions.

## Author summary

Spatial emanators are an innovative and user-friendly tool for controlling *Aedes* mosquitoes that transmit viruses such as dengue, chikungunya, and Zika. These devices combine lethal and behavioral effects on mosquitoes and are designed to passively release insecticides into the air at room temperature. These may be suitable for deployment in houses with the aim of creating "bite-free" spaces. By removing the need for conventional application methods, these devices might be rapidly deployed with minimum disruption to households. This study tested the effectiveness of emanators managed by community members versus researchers. In a trial with 588 households, emanators were replaced every three weeks across four cycles. One group of households handled installation and replacement with minimal guidance ("community-led deployment"), while researchers managed these tasks in the other group ("managed deployment"). The groups switched roles after the fourth cycle. The results showed that emanators reduced mosquito numbers by around one-third in both groups. When measuring mosquito bites, emanators reduced bites by up to 94% in the managed deployment group and up to 79% in the community-led deployment group. Importantly, households performed as effectively as researchers. This study shows that emanators can empower communities to protect themselves, making them a valuable tool during outbreaks of mosquito-borne diseases. The results are particularly encouraging given the high rates of insecticide resistance of *Ae. aegypti* in the study.

## Introduction

The yellow fever mosquito *Aedes aegypti*, the primary urban vector of dengue, Zika, and chikungunya viruses, is an opportunistic species that is highly, if not exclusively, anthropophilic, feeding preferentially on humans [1,2]. However, the main approaches used for the emergency control of *Aedes*-borne viruses (ABVs) outbreaks involve the

application of insecticides outdoors (e.g., vehicle-mounted ULV spraying) and less frequently, indoors (e.g., indoor space spraying, targeted residual treatments) across large numbers of households located at or near the home of symptomatic cases [3]. Although significant entomological impacts may result [4–6] this approach relies on considerable human resources, logistical support, and community compliance to achieve effective coverage. A major barrier to effective vector control during outbreaks is that the rapid and extensive coverage of households is challenged by the time it takes spray teams to treat interiors, the difficulty of gaining entrance, and community compliance [7,8]. Another major obstacle is that, *Ae. aegypti* has evolved resistance to many of the active ingredients used for control, particularly pyrethroids [6,9].

Spatial emanators (SE), affecting location, blood-feeding behavior and thus, mosquito–human contact, are a promising tool for controlling *Ae. aegypti* and preventing ABVs, particularly in intra-domiciliary spaces [10–12]. These repellents, as devices that contain volatile active ingredients that disperse in the air, can be deployed either as a programmatic tool by Ministries of Health (MOHs) or distributed directly to communities for home installation. Morrison et al. [13] conducted an epidemiological cluster randomized trial (CRT) in Iquitos, Peru, evaluating a SE prototype containing the active ingredient transfluthrin. The study demonstrated a 12% reduction in blood-fed female *Ae. aegypti* in treated households and a 34% decrease in dengue infections. The intervention involved insecticide-treated plastic sheets (approximately 0.5% active ingredient weight/weigh) hung from elastic lines across the roof spaces of treated households, replaced every two weeks.

A recently completed entomological randomized trial in Yucatan, Mexico, evaluated an alternative prototype of a polyurethane mesh containing metofluthrin (10% a.i. w/w) in 200 households [12]. The trial showed a significant impact, with 80–90% reductions in human landing rates and 50–60% decreases in indoor *Ae. aegypti* abundance (blood-fed mosquitoes). Furthermore, Devine et al. [12] demonstrated that metofluthrin emanators were highly effective when deployed at a rate of one device per room and replaced every three weeks, even in populations of *Ae. aegypti* with high frequencies of *kdr* alleles. Other studies evaluating SE containing these active ingredients report similar impacts and replacement schedules [14,15].

In order for SE to be effective as a vector control tool during an ABVs outbreak, they must be deployed rapidly and across large areas in locations with confirmed cases or historical transmission hotspots. Unfortunately, all research studies to date have assessed efficacy after SE were installed by a highly trained and dedicated research team leaving a gap in understanding the potential for community-led deployment to achieve comparable outcomes [12,13]. During an outbreak, one of the most critical factors for achieving widespread intervention coverage is the rapid distribution of SE to households. Whether the efficacy of SE remains high when installed directly with the active participation of the community has not been quantified. This process would need to rely on minimal guidance for installation and use, such as a brochure and a brief oral explanation.

The present study seeks to address this gap by conducting a cluster randomized control trial (cRCT) with a crossover design in the city of Merida, Mexico. This study tests the hypothesis that household members can install a prototype metofluthrin emanator and achieve entomological efficacy comparable to installations performed by an experienced vector control team. By investigating managed deployments and community-led, this research provides insights into scalable and sustainable approaches with effective innovative tools that can be delivered with community participation for reducing ABVs and other mosquito-borne disease transmissions, which is one of the IVM pillars [16].

## Materials and methods

### Ethics statement

This study was approved by the Ethics and Research Committee of the Universidad Autónoma de Yucatán (Approval No. UPI/393/2021). Written informed consent was obtained from all participants (heads of households) and each volunteer (field technician).

### Study area

This study was conducted in three suburban neighborhoods connected to Merida (Yucatan State, Mexico). A total of 588 houses participated in the study from San Lorenzo (n = 252), ACIM ("*Ampliacion Ciudad Industrial Merida*") (n = 210), and

Itzincab (n = 126) (Fig 1), similar in housing size and design, e.g., one story, brick-and-mortar homes with typically two bedrooms, one living-dining room, a kitchen, and a bathroom, characteristic of high-density low-income housing in the region as described in Vazquez-Prokopec et al. [6]. Merida is located in a subtropical environment with mean temperatures ranging from 29°C in December to 34°C in July. The rainy season occurs from May to October and overlaps with the peak dengue transmission season between July and November, although cases occur year-round [17].

### Trial design

An implementation trial was designed to compare the efficacy of a SE deployed by an operational research team (managed deployment [MD]) versus deployment by householders (community-led deployment [CD]). A three-arm, open-label entomological cluster randomised controlled trial (cRCT) with a crossover design was conducted, comparing MD and CD to a control arm with no deployment of SE.

Based on the success of a prior trial [6], the study included 42 clusters (14 per arm). Clusters were defined as entire blocks of houses, and all the clusters were selected from suburban areas within the three neighborhoods included in the study. From a total of 173 eligible blocks identified across the three neighborhoods, 42 blocks (clusters) were randomly selected using simple random sampling stratified by neighborhood, to ensure balanced allocation among arms and neighborhoods. To minimize interference between treatments caused by mosquito dispersal, clusters were non-contiguous (Fig 1).

For treatment clusters, SE were deployed in each of the 14 participating households. The trial was conducted during the ABVs transmission season (July–December 2021), a period characterized by high mosquito density. Additionally, to

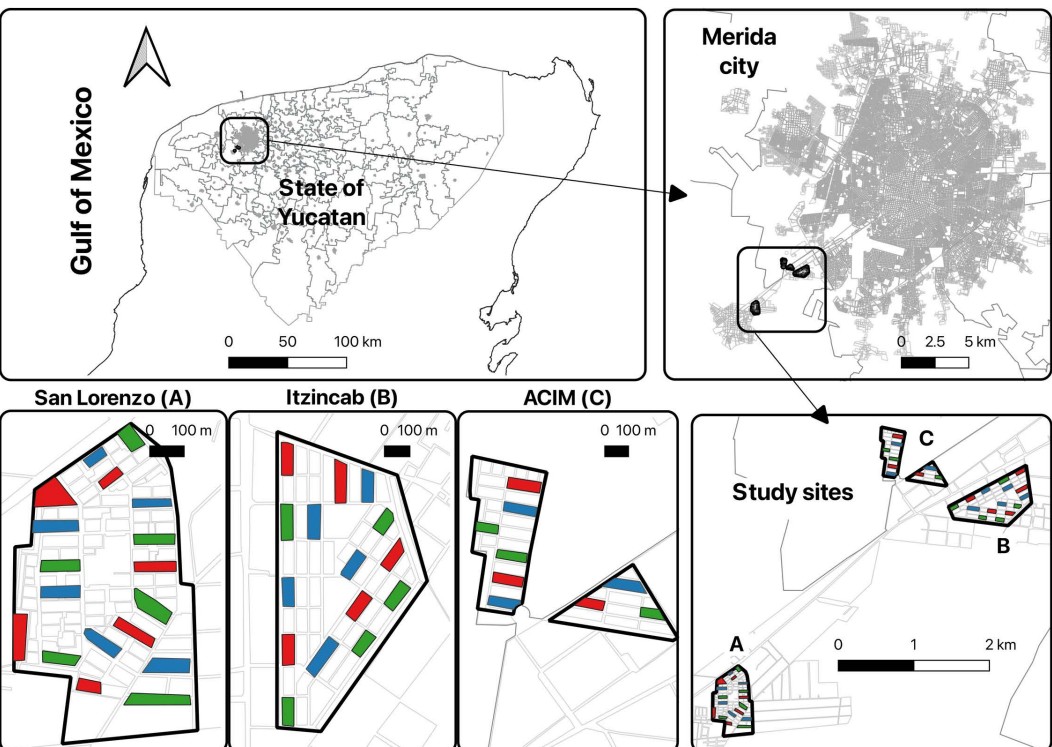

**Fig 1. Study Area.** Map of the location of the three Merida suburbs (inset): **(A)** San Lorenzo, **(B)** Itzincab, **(C)** ACIM. Clusters of control (red blocks), MD-CD (green blocks), and CD-MD (blue blocks) arms are shown. The base map was created in QGIS 3.36.1-Maidenhead (qgis.org) using layers OpenStreetMap (http://www.openstreetmap.org/), under the Open Database License (https://www.openstreetmap.org/copyright).

determine whether households that experienced deployment by an expert vector control team were more likely to deploy the emanators effectively compared to households receiving only minimal instructions, we employed a "crossover" design. After four deployment cycles (12 weeks), the two treatment arms were switched (MD to CD and CD to MD). This approach allowed households that initially observed deployment by the operational research team (MD) to later install their own devices. To evaluate whether *kdr* mutations associated with pyrethroid resistance affect the field efficacy of metofluthrin emanators, genomic DNA was extracted from individual field-caught mosquitoes. Allele-specific PCR methods were then used to detect *kdr* mutations with known functional significance in individual mosquitoes.

**Metofluthrin emanators.** The metofluthrin emanators consist of a methacrylate polymer net impregnated with 10% w/w (approximately 0.217 g) of the synthetic, volatile pyrethroid metofluthrin (Sumitomo Chemical Company Ltd., Chuo-ku, Tokyo, Japan). Various iterations of this formulation are currently registered in Australia, Singapore, Malaysia, and Thailand (e.g., Australian APVMA approval 70086/62469, Singapore NEA approval I-AmbEN/048/0829), where they are marketed as domestic consumer products for the prevention of mosquito bites indoors. The impregnated net is housed within a 95 mm x 160 mm plastic holder, designed to be hung or placed in rooms with gentle air circulation to facilitate volatilization. Strong airflows may dilute the device's efficacy. Sumitomo Corporation delivered 20,000 emanators to UADY in January 2021. This quantity was sufficient for the deployment of up to six emanators per household across 400 households, with eight replacement cycles per household (19,200 units total).

**Treatment arms.**

**Community-led deployment arm (CD):** Householders in the community-led deployment (CD) arm were responsible for installing and replacing the emanators with minimal external support. To facilitate this process, they were provided with a kit that included: a simple brochure (S1 Fig) with clear, step-by-step instructions for installation and replacement, instructional materials emphasizing ease of use, complemented by visual aids to ensure proper placement of the devices, basic tools to assist with installation and setup. A verbal briefing to reinforce the instructions and address any questions was also provided along with the kit. Additionally, householders received a reminder service via SMS text messaging to notify them when the devices needed to be replaced. Each household was supplied with five emanators (one per room), which were color-coded by replacement cycle. This color-coding system enabled householders and field observers to easily track and confirm that devices were being replaced as recommended.

**Managed deployment arm (MD):** Managed deployment involves the installation of emanators by an experienced research team following the methods described in a previous trial [12].

**Control arm:** No treatment (beyond routine vector control activities). The routine vector control program conducted by the local MOH includes truck-mounted ULV spraying of chlorpyrifos, larviciding with Methoprene, and indoor space spraying with chlorpyrifos at the premises of symptomatic cases reported to the healthcare system. Routine vector control activities occurred in both arms.

For a subset of all households, basic measures of the indoor space were recorded including total area, building materials, area of each room, and the number of doors and windows. These measures were used to guide the optimum installation (one emanator per 9–16 m2). Emanators were not installed in hallways or corridors but were installed in every other room. Emanators were hung from ceilings, above head height, to keep them clear of routine household activity, using existing fixtures (nails, hooks, light fittings) or adhesive pegs.

**Crossover of trial arms:** Crossover studies offer advantages over standard longitudinal designs. First, the impact of unrecognized variables within the study arms is minimized because each crossover household serves as its own control. Second, the statistical power of the trial is enhanced, which mitigates the effects of low mosquito densities, high non-compliance, or drop-out rates among recruited households.

Deployments were conducted over a six-month period, consisting of eight replacement cycles (each cycle lasting three weeks). Since the effect of the metofluthrin devices is demonstrable within a small number of replacement cycles, the crossover of treatment arms was initiated after the fourth replacement cycle (12 weeks post-deployment). At this point, CD

and MD arms were switched. This allowed us to test a secondary hypothesis: households that observed deployment by an expert vector control team were more likely to deploy the emanators effectively following that period of observation.

**Power and sample size calculations:** In our most recent trial [12], a two-arm randomized controlled trial (RCT) with 100 houses per arm, we achieved sufficient statistical power to detect a 60% reduction in abundance rate ratios. For the cRCT, we assessed the impact of metofluthrin emanators on the entomological endpoints in approximately 100 houses per treatment arm (i.e., 50% of the households in which emanators were installed) during two periods (before and after the crossover). Since the crossover design aims to evaluate two complementary hypotheses, we calculated power for each period independently rather than as a composite for the entire trial. Using simulations to replicate our trial design (100 Monte Carlo simulations of a cRCT involving 14 clusters, each with 7 houses, for a comparison between the control and each treatment arm) and data from our previous RCT, we determined that the cRCT would have a statistical power of 0.99 (95% CI: 0.89–0.98) to detect a difference between MD and the control. These calculations were based on an inter-cluster correlation coefficient (ICC) of 0.1. To evaluate the power to detect differences in the CD arm, we assumed that this strategy would be less effective than the managed deployment arm (a 25% reduction in efficacy, leading to an overall efficacy of 35%). Under these assumptions, we found that our trial design still has sufficient power to detect a statistically significant difference between arms (power = 0.83; 95% CI: 0.60–0.79). These calculations apply to each period of the crossover design. All calculations were performed using the *count.sim* function in the *clusterPower* package in R [18].

**Entomological endpoints.** Entomological sampling was conducted in all arms within a randomly selected subset of 7 houses per cluster (from a total of 14 houses per cluster). Sampling occurred during standard working hours (8 am–noon and 2 pm–6 pm) one week following each cycle of installation or replacement (i.e., 8 sampling weeks). The following entomological endpoints were measured: indoor *Aedes aegypti* adult abundance (including female abundance, blood-fed female abundance) as primary endpoint, and estimates of *Ae. aegypti* landing behavior, as a secondary endpoint.

**Adult indoor resting mosquitoes:** Mosquitoes were collected from all rooms within each house enrolled in the trial using Prokopack aspirators. Two field collectors aspirated mosquitoes for a total of 10 minutes per house, distributing the time evenly across all rooms. These collections were estimated to capture >75% of all resting adults indoors [19].

Collected mosquito samples were processed on the same day. Mosquito identification was carried out by expert personnel familiar with the identification of *Aedes aegypti* adults. For each mosquito, the date, house identification number, species, sex, and the presence or absence of a full or partial blood meal were recorded. Individual *Ae. aegypti* mosquitoes were preserved in vials containing 1 mL of RNAlater (Thermo Fisher Scientific, Waltham, MA, USA) and 0.1% Tween 20 (Sigma-Aldrich Co.). Over 3,000 mosquitoes were vialed and sent to QIMR Berghofer (Queensland, Australia) for screening for point mutations associated with pyrethroid resistance.

**Mosquito landing behavior:** Human landing counts [12] were conducted in 12 houses (4 per arm) selected based on high baseline entomological indices, resident willingness, and ease of access. Evaluations were performed at baseline and during replacement cycles 1, 4, 5, and 8. Experienced field workers quantified mosquito landings by sitting in one room of a selected household with both legs exposed while remaining otherwise fully protected. As mosquitoes attempted to land, the technician waved them away with their hands. This method prevented biting and avoided confounding the results through the sequential removal of landing mosquitoes during the observation period. Measurements were conducted by teams of three field workers, with each member observing in a different living space or bedroom. Each assessment lasted for 10 minutes, and the results were pooled for analysis.

**Detection of kdr alleles:** Genomic DNA extraction from field-caught mosquitoes was conducted using Extracta DNA Prep for PCR–Tissue (QuantaBio, Beverly, MA), following methods detailed in Devine et al. [12]. Briefly, individual whole mosquitoes were placed in 25 µL of extraction reagent, incubated at 95˚C for 20 minutes, and then cooled to room temperature. Subsequently, 25 µL of stabilization buffer was added, and samples were stored at -20˚C until use. Allele-specific PCR methods were employed to detect *kdr* mutations with known functions. Genotyping was performed using a CFX-96 RT-PCR system (Bio-Rad, Hercules, CA) under specific cycling and melt curve conditions. The primers used were

adopted from Saavedra-Rodriguez et al. [20] for V1016I, Yanola et al. [21] for F1534C, and Saavedra-Rodriguez et al. [22] for V410L. PCR reagents and conditions were based on Deming et al. [23] and Saavedra-Rodriguez et al. [20] for V1016I, Deming et al. [23] for F1534C, and Saavedra-Rodriguez et al. [22] for V410L.

**Satisfaction and emanator status.** Structured questionnaires on satisfaction, perceived efficacy and correct use of ES, were administered in every household where emanators were deployed. Surveys were conducted at two time points: during the 12th week of implementation (prior to the treatment switch), and during the 24th week of implementation (end of the study).

The questionnaires were identical for both intervention arms that received emanators. Satisfaction levels were categorized from lowest to highest as: "not satisfied," "somewhat satisfied," and "very satisfied." Two aspects of satisfaction were evaluated: a) overall satisfaction with the presence of emanators in the household, and b) satisfaction with the installation and replacement process.

**Analysis.** *Ae. aegypti* adult indices were calculated for each sampling date and compared between treatments and over time. To evaluate the entomological impact of the emanators, a crossover analysis was conducted using a negative binomial GLMM. This analysis included fixed effects for treatment arm and "carryforward" effect (tracking whether a household experienced a treatment switch) and random intercepts for house ID and survey time points. Based on the coefficients of the best-fitting model, determined by the lowest AIC, we estimated the entomological impact of the emanators after accounting for the switch in arms, the heterogeneity of the households, and the multiple survey time points.

Values of each endpoint during all sampling periods were compared using generalized linear mixed effects models (GLMM) nested at the cluster (level 1) and house (level 2) levels. This nesting structure explicitly accounted for the clustering of observations at the household level, recognizing that repeated measurements within households are not statistically independent. Models were used to calculate incidence rate ratios (IRR), using control houses within their respective clusters and blocks as the unit of comparison. The operational efficacy of the intervention was calculated as $E = (1 - IRR) \times 100$. This measure, ranging from 0 to 100, represents the percent reduction in mosquito abundance in treated houses relative to control houses. Covariates such as seasonality and geographic variation were included in the statistical models to account for potential confounders. While the inter-cluster correlation coefficient (ICC) of 0.1 was described earlier, it was specifically selected based on prior studies in similar settings to reflect moderate clustering of mosquito populations.

All models were implemented using the lme4 package [24] within the R software platform (R Core Team, 2022). Using the same software, pooled human landing behavior data were analyzed with GLMM employing a Poisson link function and a random effect associated with each house identifier. This analysis aimed to a) quantify differences between treatment and control arms, b) plot the model-predicted number of attempted landings using the *ggeffects* package in R [25].

## Results

### Impact on *Ae. aegypti* density indoors

A total of 3,323 *Ae. aegypti* adults were collected indoors using Prokopack aspirators during the trial (sex ratio, 1.15:1.0 F:M). The control, CD, and MD arms yielded 1,366, 1,025, and 932 individuals, respectively. Other mosquito species collected included 913 *Culex quinquefasciatus* (513, 168, and 215 individuals in the control, CD, and MD arms, respectively) and 12 *Aedes albopictus* adults.

Fig 2 illustrates the mean number of adults and blood-fed females per house at each survey point and study arm. The initial deployment of metofluthrin emanators resulted in a significant reduction in mean *Ae. aegypti* density in both treatment arms (CD and MD) compared to the control arm, with this effect lasting until the third survey date, prior to the treatment switch (Fig 2A and 2B). Notably, the reductions in mosquito abundance were more pronounced during the first third cycles of deployment, suggesting that initial placement may be critical for maximizing efficacy.

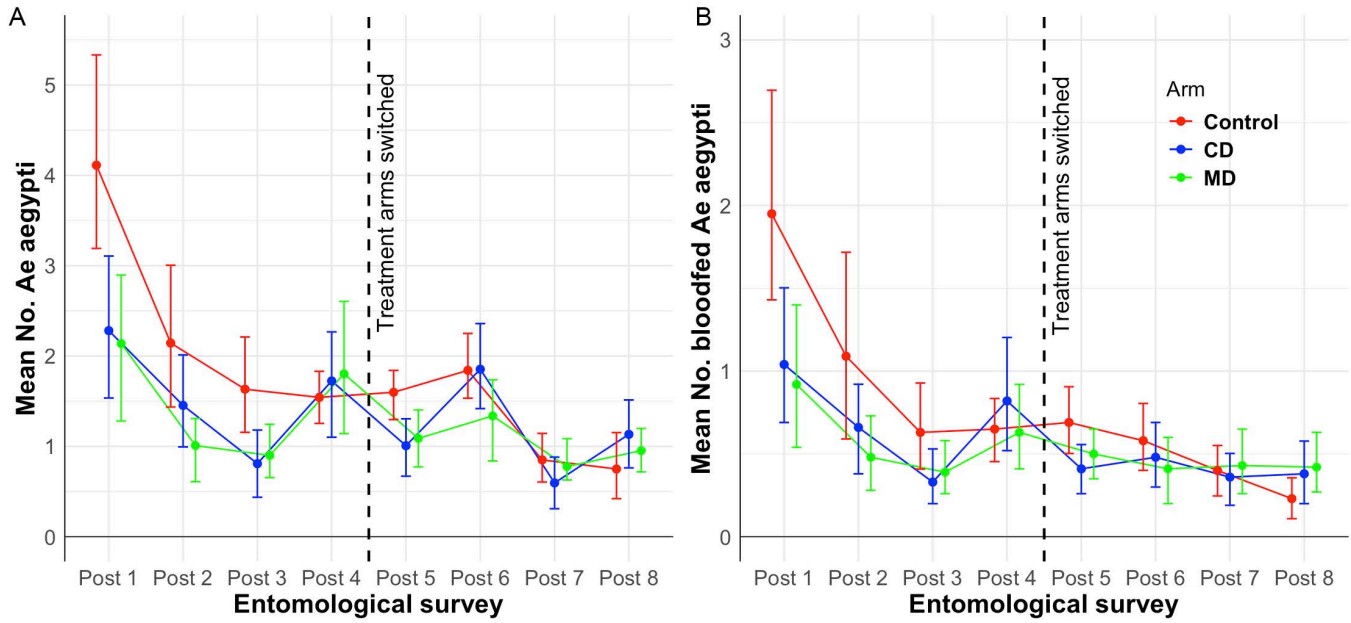

**Fig 2. Entomological impact on the abundance of *Ae. aegypti*.** (A) Mean number (error bars indicate 95% CI) of *Ae. aegypti* adults per house and per treatment for each entomological survey before and after the switch of treatment arms (indicated as a vertical dashed line). (B) Mean number of blood-fed *Ae. aegypti* females per house per survey date and treatment. Note that during the last 8 weeks of the trial, mosquito abundance was very low.

In the second study period (after switching treatments), reductions in *Ae. aegypti* indices were less pronounced. Densities ranged from 0.8 to 1.7 adults per house in the control arm, 0.35 to 1.7 in the CD arm, and 0.8 to 1.1 in the MD arm (Fig 2A). A similar pattern was observed for blood-fed female *Ae. aegypti* (Fig 2B).

Table 1 show the results of crossover analysis using a negative binomial GLMM including as fixed effects the treatment arm and the "carryforward" effect, and as random intercepts the house ID and survey time points (Table 1). Model-based estimates were presented only for the adult abundance endpoint, as this variable demonstrated sufficient data stability and model fit to support reliable inference. The best-fitting model (lowest AIC) included arm and "carryforward" effect as fixed effects.

The model revealed that both CD and MD deployments of metofluthrin emanators significantly reduced the number of *Ae. aegypti* adults per house compared to the control, even after accounting for repeated measures and the switch between treatment arms at survey 5. Interestingly, no "carryforward" effect was detected, indicating that the initial

**Table 1. Model selection.**

| Model | Fixed Effects | Random Effects | AIC[1] |
|---|---|---|---|
| 1 | Arm, Carryforward | House ID, Visit ID | 7409 |
| 2 | Arm, Carryforward, Emanator Status | House ID, Visit ID | 7415 |
| 3 | Arm, Carryforward | Cluster ID, Visit ID | 7470 |
| 4 | Arm, Carryforward | Cluster ID, House ID | 7494 |
| 5 | Arm, Carryforward, Switch | House ID | 7494 |

Comparison of prediction errors for models of the entomological evaluation adult abundance using a negative binomial GLMM.

[1]AIC (Akaike Information Criterion) is an estimator of prediction error and the relative quality of statistical models for a given set of data. It provides a means for model selection.

deployment method—whether by the community or the research team—did not influence the entomological impact following the treatment switch (Table 2).

Based on the coefficients of the best-fitting model GLMM, we estimated the entomological impact of the emanators after accounting for the switch in arms, the heterogeneity of the households, and the multiple survey time points. Fig 3 clearly shows a significant reduction in the number of adult *Ae. aegypti* indoors when emanators are present, and little difference between CD or MD arms. Using the coefficients of the best-fitting model we estimated that the intervention reduced the number of *Ae. aegypti* per house by 32.7% (95%CI = 16.2-46.0%) for the CD arm and 36.8% (21.1-49.3%) for the MD arm.

Table 4 provides the estimated IRR for attempted landings for each treatment arm relative to the control. Values below 1 indicate reductions in mosquito landings compared to the baseline. Across all survey dates, HLCs in both treatment arms were significantly lower than in the control arm. Using an intervention efficacy equation (%Eff = 1 − IRR*100), the MD arm consistently exhibited higher efficacy (74–94%) in reducing mosquito landings compared to the CD arm (35–79%) (Fig 4).

### Detection of *kdr* alleles

The metofluthrin emanators demonstrated efficacy against a mosquito population with very high frequencies of pyrethroid-resistant alleles. All three alleles (V1016I, F1534C, and V410L) associated with conventional pyrethroid resistance were present at high frequencies in the trial site population. The triple mutant homozygote accounted for over 50% of mosquitoes, and nearly 100% of individuals were homozygous for the F1534C mutation. There was no evidence of a change in resistant allele frequencies between treatment arms (Fig 5A) or over time (Fig 5B).

**Table 2. Results from the optimal GLM model.**

| Parameter[1] | Estimate | Std. Error | z-value | P-value |
|---|---|---|---|---|
| (Intercept) | 0.2752 | 0.1345 | 2.046 | 0.0408 |
| Arm = Community (CD) | -0.3958 | 0.1118 | -3.54 | 0.0004 |
| Arm = Managed (MD) | -0.4582 | 0.1127 | -4.067 | <0.0001 |
| Carryforward = MD to CD | 0.1759 | 0.1521 | 1.157 | 0.2473 |
| Carryforward = CD to MD | 0.1985 | 0.1498 | 1.325 | 0.1852 |
| **Random Effects** | | | | |
| **Parameter** | **Type** | **Variance** | **Std.Dev.** | |
| House ID | (Intercept) | 0.296 | 0.5441 | |
| Entomological Survey | (Intercept) | 0.1108 | 0.3329 | |

[1]Letters indicate the following treatment arms: community deployment (CD), and managed deployment (MD).

A further comparison of the log-scale model estimates confirmed that the only significant differences in entomological impact were observed between each treatment arm (CD and MD) and the control, with no significant difference between the CD and MD arms (Table 3).

**Table 3. GLMM model comparisons of treatment and control arms: impact of metofluthrin emanators vs number of *Ae aegypti* per house.**

| Contrasts | estimate | SE | z.ratio | P-value |
|---|---|---|---|---|
| **Control-Community (CD)** | 0.3958 | 0.112 | 3.54 | 0.0012 |
| **Control-Managed (MD)** | 0.4582 | 0.113 | 4.067 | 0.0001 |
| **Community (CD) – Managed (MD)** | 0.0624 | 0.116 | 0.538 | 0.8526 |

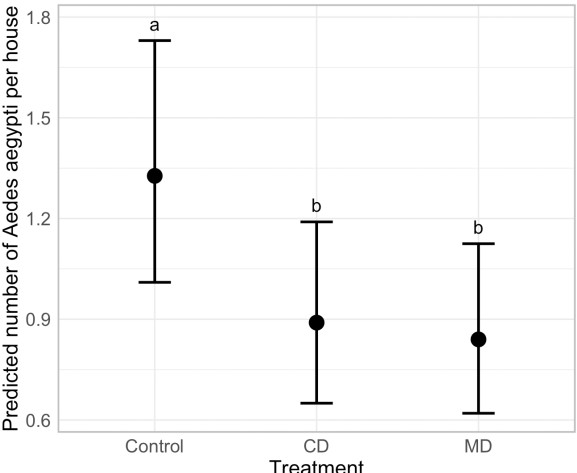

**Fig 3. Estimated number of *Ae. aegypti* from the best fitting GLMM.** Data presented as predicted means ± 95% CI. Different letters indicate statistically significant differences (P < 0.05).

**Table 4. Predicted Incidence Risk Ratio (IRR) for attempted landings.**
Negative-binomial GLMM results comparing the mosquito landings, between the treatment arm (CD and MD) and the control. The efficacy of intervention (%Efficacy = 1-IRR*100) is also shown.

| Survey | Arm | Est. HLC (95% CI) | Est. IRR | % Efficacy (95% CI) |
|---|---|---|---|---|
| Baseline | Control | 3.2 (2.3-4.5) | | |
| | CD | 6.9 (5.4-8.7) | 2.16 | – |
| | MD | 11.7 (9.8-14) | 3.66 | – |
| Post 1 | Control | 9.9 (8.1-12.1) | | |
| | CD | 6.44 (5-8.3) | 0.65 | 35 (10-53) |
| | MD | 2.6 (1.8-3.8) | 0.26 | 74 (60-83) |
| Post 4 | Control | 7.2 (5.7-9.1) | | |
| | CD | 1.5 (0.9-2.5) | 0.21 | 79 (65-89) |
| | MD | 1 (0.54-1.9) | 0.14 | 86 (74-93) |
| Post 5 | Control | 11.4 (9.5-13.7) | | |
| | CD | 3.7 (2.7-5.1) | 0.32 | 68 (53-78) |
| | MD | 0.7 (0.3-1.5) | 0.06 | 94 (88-97) |
| Post 8 | Control | 7.9 (6.3-9.8) | 7.90 | |
| | CD | 2.7 (1.8-3.9) | 0.34 | 66 (48-78) |
| | MD | 1.9 (1.2-3) | 0.24 | 76 (61-86) |

## Evaluation of satisfaction and use of the SE

A total of 749 surveys were administered to evaluate user satisfaction with the emanators. Of these, 388 surveys (194 in each arm that received emanators) were conducted at the end of the first phase of implementation, and 361 were conducted at the end of the second phase (189 in the MD arm and 172 in the CD arm); overall satisfaction with emanator use in the household was high, with the "very satisfied" category consistently exceeding 95% across both implementation models (S1 Table).

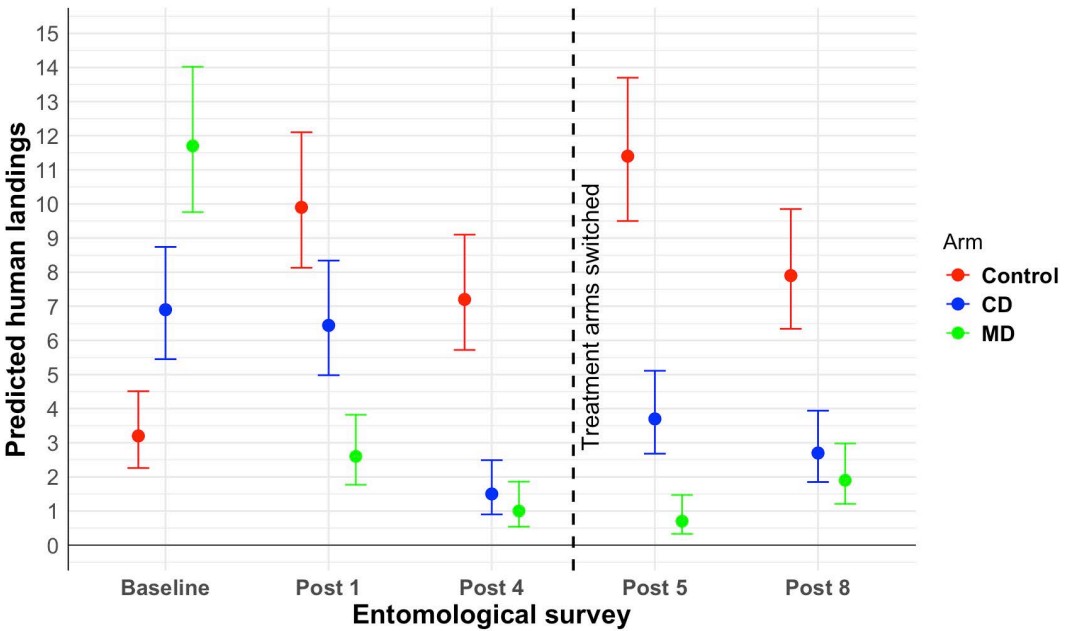

**Fig 4. Predicted number of attempted landings by study arm.** Predictions were obtained from a GLMM with a Poisson link function. The vertical black line shows the separation of surveys before and after the switch in study arms.

Across the eight replacement cycles, the percentage of households with all emanators correctly installed was 82.2% and 75.4% in those assigned to the MD and CD arm, respectively. Regarding the effect of arm crossover on installation accuracy, households that initially received the CD showed a 25.8% increase in correct installation rates after switching to the MD model. Conversely, households that began with the MD showed a 13.1% increase when switched to the CD model.

Incorrect installation of emanators was often attributed to structural characteristics of the house, such as ceiling type (e.g., textured "tirol" finishes) or excessive humidity. In cases where emanators fell, participants typically attempted to reinstall them on their own; however, some chose to wait for the research team's visit to report the incident and stored the fallen devices in the meantime.

## Discussion

The deployment of metofluthrin emanators in Merida resulted in significant reductions in the number of *Ae. aegypti* per house and the number of attempted landings. Few studies have measured behavioral impacts on host-seeking behaviors in the field, even though the novelty of the SE paradigm is largely about that behavioral effect. Entomological impact was consistent with other previously reported results [12,13], adding more evidence and confirming that emanators and/or spatial repellent formulations like these, can have an important impact on *Ae. aegypti* population densities and human-vector contact indoors. Importantly, this intervention has consistently demonstrated efficacy in resistant mosquito population [12], reinforcing its potential utility for public health vector control responses against ABVs.

Although human landing catch (HLC) data were collected from a small number of houses (n = 4 per arm), this approach followed established protocols from previous ES studies [12] and was chosen for logistical and ethical reasons. Repeated measures within these households over time allowed for consistent assessment of host-seeking behavior. Despite the limited sample size, the observed reductions in mosquito landings align with trends in adult mosquito abundance, providing important complementary evidence that metofluthrin emanators reduce human-vector contact indoors.

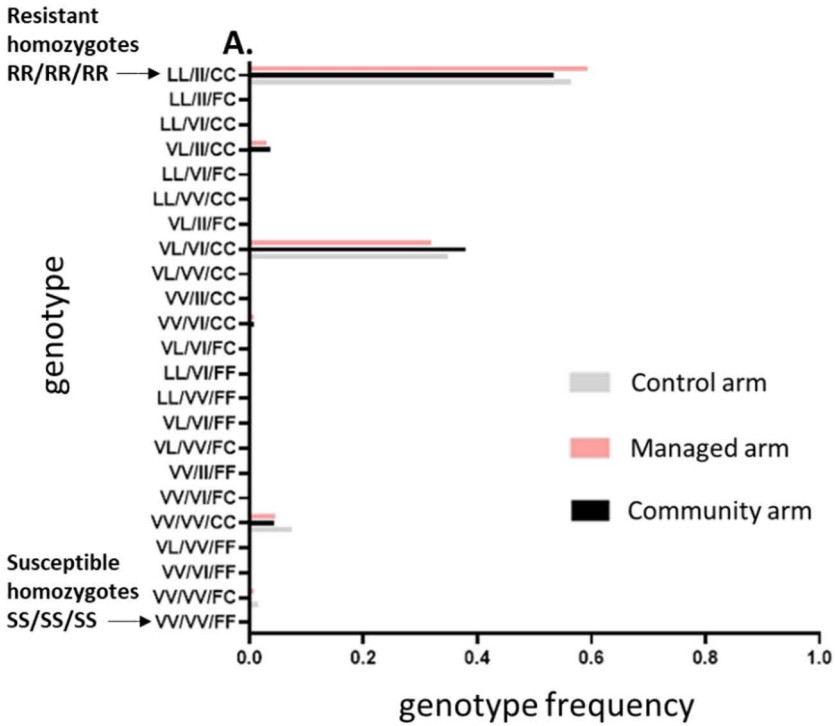

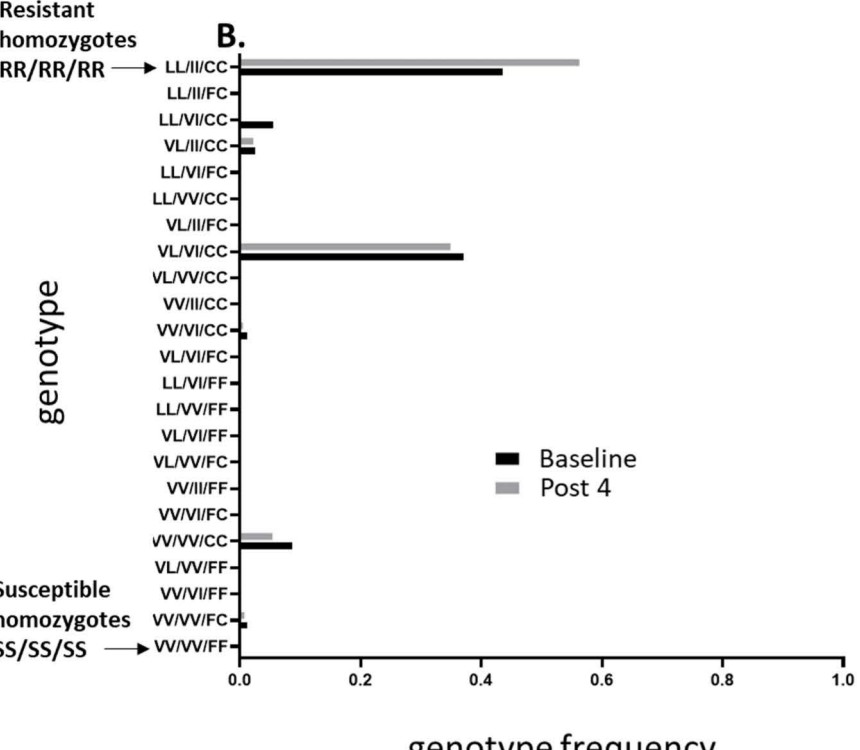

**Fig 5. Comparison of Resistant Allele Frequencies: Resistant homozygous forms (RR) are represented as LL, II, and CC, respectively, at the top of the graph, while sensitive homozygous forms are denoted as VV, VV, and FF. (A)** Displays a comparison of resistant allele frequencies across the study arms. **(B)** Shows all arms combined, comparing resistant allele frequencies between the baseline and the 4th replacement cycle.

Despite the high frequencies of pyrethroid-resistant alleles in the mosquito population, reductions in landing activity (60–80%) and abundance (30–40%) were comparable to those reported for SE products in previous studies [12,13]. For example, Morrison et al. [13] found that a modest 12% reduction in blood-fed *Ae. aegypti* led to a > 30% decrease in dengue transmission. The entomological impacts observed in the current trial suggest that a similar epidemiological impact is likely. This highlights the potential of SE not only as a preventive measure but also as a tool for outbreak containment. Future studies should aim to directly link entomological outcomes with epidemiological metrics, such as disease incidence or transmission rates.

The consistent impacts across treatment arms, demonstrated that communities could implement SE as effectively as an experienced vector control team, even with minimal instruction or support. This is the first study to show that SE products can be successfully deployed by the community and a feasible approach. Moreover, the crossover design allowed us to investigate community "learning" from the research team. Results indicated that, following minimal guidance (see S1 Fig for the brochure provided to communities), the placement of emanators by the community achieved a similar entomological impact to that of a more structured deployment by the research team. This effectiveness may also stem from the prototype emanator used, which required only one unit per room. Also, in our study, "emanator status" was neither significant nor included in the top model, indicating that placement accuracy did not affect the observed entomological reductions.

Community-based mosquito control methods are effective when there is active participation and education within the community [26–28]. While it has been hypothesized that observing an expert installation team might improve the efficiency and accuracy of community deployment, our findings provided no evidence to support this. If metofluthrin emanators are correctly installed, they appear to have sufficient intrinsic efficacy to reduce the density and biting behavior of *Ae. aegypti* indoors. The absence of a "carryforward" effect underscores the practicality of community-led deployment. This finding indicates that prior exposure to expert deployment is not a prerequisite for effective implementation, making the approach highly adaptable for rapid responses during outbreaks.

We identified that the high acceptability of the strategy was driven by the perceived high risk of dengue infection in the participants of both the CD and MD strategies (reasons for participation were obtained through satisfaction surveys after the 4th visit to install/replace emanators). However, the less intrusive nature of the CD strategy resulted in a clear community preference for this approach, highlighting its potential as an effective alternative to traditional vector control methods. Householders appreciated the ability to install the emanators at their convenience and in their preferred locations (following the provided guide, S1 Fig), making the intervention more acceptable than having a research team enter their homes for installation. This preference suggests that a CD approach to deploying emanators could significantly increase the coverage of insecticidal control campaigns. By reducing the burden on operational teams, this approach offers a scalable model for vector control during outbreaks, especially in resource-limited settings; such an approach can be tailored to different epidemiological scenarios, particularly during outbreaks.

While our findings are robust, the study was limited to a specific geographic and seasonal context. The results may differ in areas with varying mosquito behaviors, housing structures, cultural perceptions, or compliance levels. Additional research is required to assess long-term acceptance, sustained efficacy, and potential barriers to implementation in diverse settings.

In conclusion, and according to our results, community-based deployment of spatial repellents presents a feasible and effective alternative to traditional vector control methods. By increasing the coverage and flexibility of insecticidal interventions, this approach could play a vital role in enhancing public health responses to vector-borne disease outbreaks.

## Supporting information

**S1 Fig. Brochure provided to communities containing clear, step-by-step instructions for the proper installation and timely replacement of the emanators.**
(PDF)

**S1 Table. Overall user satisfaction by implementation model before and after the crossover.**
(DOCX)

## Acknowledgments

We thank the residents of the study localities for their cooperation and enthusiasm and the Innovative Vector Control Consortium (IVCC) team for their support. The emanators were provided by Sumitomo Chemical Company.

## Author contributions

**Conceptualization:** Gregor Devine, Gonzalo Vazquez-Prokopec, Pablo Manrique-Saide.

**Formal analysis:** Azael Che-Mendoza, Guillermo Guillermo-May, Oscar D Kirstein, Gonzalo Vazquez-Prokopec.

**Funding acquisition:** Gregor Devine.

**Investigation:** Azael Che-Mendoza, Guillermo Guillermo-May, Aylin Chi-Ku, Norma Pavía-Ruz, Anuar Medina-Barreiro, Gabriela González-Olvera.

**Project administration:** Gregor Devine, Gonzalo Vazquez-Prokopec, Pablo Manrique-Saide.

**Writing – original draft:** Azael Che-Mendoza, Guillermo Guillermo-May, Oscar D Kirstein, Gonzalo Vazquez-Prokopec, Pablo Manrique-Saide.

**Writing – review & editing:** Aylin Chi-Ku, Norma Pavía-Ruz, Anuar Medina-Barreiro, Gabriela González-Olvera, Gregor Devine.

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
