## [Decision Letter · Decision Letter 0]

24 Mar 2025

Community deployment of metofluthrin emanators to control indoor Aedes aegypti: efficacy results from a crossover trial in Yucatan, Mexico.

Dear Dr. Manrique-Saide,

Thank you for submitting your manuscript to PLOS Neglected Tropical Diseases. After careful consideration, we feel that it has merit but does not fully meet PLOS Neglected Tropical Diseases's publication criteria as it currently stands. Therefore, we invite you to submit a revised version of the manuscript that addresses the points raised during the review process.

Please submit your revised manuscript within 60 days May 23 2025 11:59PM. If you will need more time than this to complete your revisions, please reply to this message or contact the journal office at plosntds@plos.org. Please include the following items when submitting your revised manuscript:

We look forward to receiving your revised manuscript.

Kind regards,

Ran Wang, M.D.

Academic Editor

Paul Mireji

Section Editor

Shaden Kamhawi

co-Editor-in-Chief

Paul Brindley

co-Editor-in-Chief

**Journal Requirements:**

1) We noticed that you used the phrase 'data not shown' in the manuscript. We do not allow these references, as the PLOS data access policy requires that all data be either published with the manuscript or made available in a publicly accessible database. Please amend the supplementary material to include the referenced data or remove the references.

- ® on pages: 14, and 16.

3) Some material included in your submission may be copyrighted. According to PLOSu2019s copyright policy, authors who use figures or other material (e.g., graphics, clipart, maps) from another author or copyright holder must demonstrate or obtain permission to publish this material under the Creative Commons Attribution 4.0 International (CC BY 4.0) License used by PLOS journals. Please closely review the details of PLOSu2019s copyright requirements here: PLOS Licenses and Copyright. If you need to request permissions from a copyright holder, you may use PLOS's Copyright Content Permission form.

Potential Copyright Issues:

i) Please confirm (a) that you are the photographer of S1, or (b) provide written permission from the photographer to publish the photo(s) under our CC BY 4.0 license.

ii) Figure S1. Please confirm whether you drew the images / clip-art within the figure panels by hand. If you did not draw the images, please provide (a) a link to the source of the images or icons and their license / terms of use; or (b) written permission from the copyright holder to publish the images or icons under our CC BY 4.0 license. Alternatively, you may replace the images with open source alternatives. See these open source resources you may use to replace images / clip-art:

4) Thank you for stating "Yes - all data are fully available without restriction." Please note that your Data Availability Statement is currently missing [the repository name and/or the DOI/accession number of each dataset OR a direct link to access each database].  Please provide a complete Data Availability Statement in the submission form, ensuring you include all necessary access information or a reason for why you are unable to make your data freely accessible. If your research concerns only data provided within your submission, please write "All data are in the manuscript and/or supporting information files" as your Data Availability Statement.

**Comments to the Authors:**

**Please note that one of the reviews is uploaded as an attachment.**

**Reviewers' Comments:**

Reviewer's Responses to Questions

**Key Review Criteria Required for Acceptance?**

**Methods**

-Are the objectives of the study clearly articulated with a clear testable hypothesis stated?

-Is the study design appropriate to address the stated objectives?

-Is the population clearly described and appropriate for the hypothesis being tested?

-Is the sample size sufficient to ensure adequate power to address the hypothesis being tested?

-Were correct statistical analysis used to support conclusions?

-Are there concerns about ethical or regulatory requirements being met?

Reviewer #1: Were there any source reduction/environmental control/hygiene steps as part of the routine vector control? Many countries start this first and continue it throughout seasonal and outbreak management.

Were there any ethics review documents you should proffer for the human landing catches or for volunteer participation in trials?

Reviewer #2: Keys for identification of the mosquitoes are missing

Reviewer #3: 1. It should be stated explicitly in the Methods and the abstract that the trial design was non-blinded/open-label

2. The description of the units of randomisation and treatment assignment at lines 169-177 on p10 is not clear enough in several aspects:

2a. This sentence ‘Clusters were defined as entire city blocks randomly selected from suburban blocks within the three neighborhoods of study’ is confusing. The distinction between ‘city blocks’ and ‘suburban blocks’ is not clear, nor is it clear how the unit of ‘block’ differs from ‘cluster’. Please state how many eligible city blocks there were in total from which this subset of 42 clusters was selected and what method was used for random selection of those 42 clusters for inclusion in the trial - e.g. probability proportionate to size? Was the selection stratified by the 3 neighbourhoods?

2b. Among the 42 clusters, it looks from Fig 1 like assignment to the three study arms was randomised within each of the 3 neighborhoods because the cluster numbers per arm are exactly balanced within each neighborhood - if so this stratified randomisation needs to be stated in the methods.

2c. The unit of randomization is the cluster, but the unit of intervention is the household and it is not clear what proportion of total houses in each cluster (and overall in each study arm) received a spatial emanator. Line 175 states “For treatment clusters, the goal was to achieve full block coverage with SE.” Does this mean the goal was to deploy an SE in every household in each intervention cluster, or if not what does ‘full block coverage’ mean? It is stated further on (p13) that each cluster included 14 participating houses - please include this information with the description of treatment assignment on page 10, and state the total number of houses in each cluster and arm and how the participating 14 houses were selected.

3. Regarding the Entomological endpoints (page 14):

3a. Please state which were the pre-defined primary and secondary endpoints for evaluating entomological efficacy: total Ae. aegypti; female; blood-fed? And report results for all of the primary and secondary endpoints (see comment 2 re Results).

3b. Why was entomological sampling only done in 50% of houses? This means that the effective sample size of the trial for the purpose of the analyses was only 7 houses per cluster x 14 clusters, and not 14 houses x 14 clusters as stated in the power calculations - as the remaining half of the houses contributed no information on entomological outcomes, if I understand the methods correctly.

3c. Please clarify whether entomological sampling was conducted ‘once a week following each cycle of installation’ as stated at line 267 (i.e. 24 sampling weeks) or whether this should read ‘one week’ (i.e. 8 sampling weeks, as suggested by Figure 2.

3d. I have particular concerns about the analyses of human landing catch data as a secondary endpoint, as these data were only collected in a total of 12 houses (4 per arm), repeated at 4 timepoints. No information is provided about how those 4 houses per arm (of 196 total per arm) were selected, and as such it is not possible to draw robust conclusions from these data.

4. Regarding the description of the GLMM statistical analysis (page 16)

4a. The terminology regarding clusters (level 1) and block (level 2) is confusing. What does a block correspond to here? Clustering in entomological indices at the household level should also be accounted for in the GLMM analysis, since the mosquito sampling was repeated over time at the same households and those repeated observations are not statistically independent.

4b. In fact, the text in the Results section indicates that models did include House ID as a random effect, but this is at odds with the model described at lines 311-2 of Methods. Please clarify and strengthen the description of the statistical modelling methods used.

5. No ethical considerations are included in the methods section, regarding review and approval of the study protocol by an IRB or regarding informed consent obtained from trial participants. This information must be included

**Results**

-Does the analysis presented match the analysis plan?

-Are the results clearly and completely presented?

-Are the figures (Tables, Images) of sufficient quality for clarity?

Reviewer #1: Any thoughts about the up and down nature of the numbers in spite of treatment? Is this the treatment wearing off and not quite lasting to the end of the designed interval?

Otherwise, this is really excellent. Data really can’t be clearer than that for this type of study.

Reviewer #2: The data is well analyzed.

Reviewer #3: 1. The description of the crossover analysis at lines 356-60 belongs in the methods section and, as above, is inconsistent with the description of the statistical model at lines 311-2 of Methods.

2. Figure 2 shows results for number of total Aegypti and number of blood-fed females per house, while the results of statistical models are only reported for the total Aegypti endpoint and not blood-fed females; as per my comment on the Methods, please report results for each of the pre-defined primary and secondary endpoints.

3. In reporting the intervention effect on p17, the authors state “...with this effect lasting until the fourth survey date”, but Figure 2 shows no difference between CD or MD groups vs Control in the Post 4 survey.

4. In addition to my comment on the Methods for selecting houses for human landing catches, it is not clear from the data presented in Table 4, and the statistical methods described at lines 321-24, how the efficacy of the intervention was estimated from the human landing catch data.

4a. Please report the absolute HLC data (e.g. mean/median and range of attempted landings per house and per time point) as well as the model output shown in Table 4

4b. Clustering of observations at the household level should be accounted for in this analysis - was it?

4c. The IRR value for the reference group (Control arm) would usually be 1.0; I don’t understand the values of IRR reported for the ‘intercept’ arm in Table 4.

**Conclusions**

-Are the conclusions supported by the data presented?

-Are the limitations of analysis clearly described?

-Do the authors discuss how these data can be helpful to advance our understanding of the topic under study?

-Is public health relevance addressed?

Reviewer #1: You should accentuate the resistance argument more. Did those prior studies also monitor their resistance and show the 30%+ effect despite the resistance? If so, make that clear that this is consistently working in resistant populations. If not, then highlight that your study took that extra step and it may infer that those other successful trials also experienced severe resistance.

It is rare that a tool this compatible with untrained users makes its way into vector control.

Reviewer #2: satisfactory

Reviewer #3: 1. The statement in the first line of the discussion that the intervention resulted in significant reductions in Ae. aegypti females/blood-fed females and the number of attempted landings is not supported by the data presented

1a. As per comments above, data in Figure 2 shows only total Ae. aegypti and blood-fed females, not Ae. aegypti female counts. And the statistical model results presented only include total Ae. aegypti, not females or blood-fed females. Results must be shown for pre-defined endpoints.

1b. The very small number of houses in which HLC data was collected; the lack of explanation of how those houses were selected; the lack of reporting of the raw data on HLC results; and the apparent lack of accounting in the statistical analysis for the clustering of data within (the very small number of) households together make the HLC very unconvincing. The above details should be reported, and the conclusions drawn from these data should be tempered to explicitly acknowledge the limitations of small sample size and (presumably) large inter- and intra-household variability in attempted landings

2. ‘Emanator status’ is referred to in the discussion, but there is no mention in the methods section about how, when or by whom placement accuracy was assessed.

3. The authors introduce for the first time in the discussion section (lines 472-84) the information that an acceptability/satisfaction survey was conducted among trial participants to understand community preferences. Please include the acceptability survey methods and results in the paper; this cannot be introduced in the discussion section without any data shown. Given that the study was focused on evaluating community deployment of spatial emanators, it seems very odd to have excluded this qualitative data from the paper.

**Editorial and Data Presentation Modifications?**

Reviewer #1: Minor revision

Reviewer #2: Minor revision

Reviewer #3: Minor comments:

1. Line 110: spell out the abbreviation ‘a.i. w/w’ at first use

2. Lines 219-222: specify whether the routine vector control activities also continued in the intervention arms

**Summary and General Comments**

Reviewer #1: This work is direct, concise, and strong. This is a fantastic paper that is laid out clearly and has a valuable message for scalability of an alternative outbreak response paradigm.

Just a couple things:

Key Words:

All of your key words are redundant with your title. Consider other options such as intervention, mass deployment, vector management, spatial repellent, pyrethroid, community behavior, bite prevention, etc.

Author Summary:

Line 72: Vectors do not transmit diseases. They transmit pathogens, the causal agents of disease.

After reading the rest of the paper, I think you should have in both the abstract and author summary a pointed statement that these were highly pyrethroid resistant mosquito populations and the intervention was significant in spite of it.

Introduction:

Line 89: the common name is the yellow fever mosquito.

Line 90: These citations aren’t really about these aspects of their biology. Endophilic is acceptable as a descriptor, though consider saying anthropophilic to be more cautious of the wildly varying and cryptic larval sources they utilize all over the world. I would not describe them as endophagic; they don’t prefer to bite indoors at the exclusion of other environments. They are opportunistic, biting at many times of day and wherever you happen to encounter them.

Reviewer #2: It is an interesting study. However, I have three concerns: (1) The identification of the mosquito species is not presented. (2) The human landing catch method for Aedes aegypti requires ethical approval, but this is not mentioned. (3) The year and date of the study are missing.

Reviewer #3: The authors report the outcomes of a three arm cluster randomized controlled trial of the entomological efficacy of metofluthrin spatial emanators in reducing indoor Ae. aegypti abundance and biting behavior in a Mexican city. The study was designed specifically to test the hypothesis that entomological efficacy is comparable when the emanators are installed by community members with limited instruction, vs the managed approach to deployments that has been used in the previous trials that have reported entomological efficacy of this intervention. As outlined in my specific comments above, I have considerable concerns with the lack of clarity in describing the approach used for cluster selection and treatment assignment; the incomplete reporting of all entomological outcomes; and the fact that the effective sample size was half of what is stated in the methods because the entomological endpoints appear to have been measured in only 50% of households enrolled in the trial; and with the sampling approach, analysis and reporting of the HLC data, all of which need to be addressed in a revised manuscript.

PLOS authors have the option to publish the peer review history of their article (what does this mean? ). If published, this will include your full peer review and any attached files.

**Do you want your identity to be public for this peer review?** For information about this choice, including consent withdrawal, please see our Privacy Policy .

Reviewer #1: No

Reviewer #2: No

Reviewer #3: No

**Figure resubmission:**

**Reproducibility:**



---

## [Decision Letter · Decision Letter 1]

23 Jul 2025

Community deployment of metofluthrin emanators to control indoor Aedes aegypti: efficacy results from a crossover trial in Yucatan, Mexico.

Dear Dr. Manrique-Saide,

Thank you for submitting your manuscript to PLOS Neglected Tropical Diseases. After careful consideration, we feel that it has merit but does not fully meet PLOS Neglected Tropical Diseases's publication criteria as it currently stands. Therefore, we invite you to submit a revised version of the manuscript that addresses the points raised during the review process.

Please submit your revised manuscript within 30 days Aug 22 2025 11:59PM. If you will need more time than this to complete your revisions, please reply to this message or contact the journal office at plosntds@plos.org. Please include the following items when submitting your revised manuscript:

* A rebuttal letter that responds to each point raised by the editor and reviewer(s). You should upload this letter as a separate file labeled 'Response to Reviewers '. This file does not need to include responses to any formatting updates and technical items listed in the 'Journal Requirements' section below.

* A marked-up copy of your manuscript that highlights changes made to the original version. You should upload this as a separate file labeled 'Revised Manuscript with Track Changes '.

* An unmarked version of your revised paper without tracked changes. You should upload this as a separate file labeled 'Manuscript '.

We look forward to receiving your revised manuscript.

Kind regards,

Ran Wang, M.D.

Academic Editor

Paul Mireji

Section Editor

Shaden Kamhawi

co-Editor-in-Chief

Paul Brindley

co-Editor-in-Chief

**Reviewers' comments:**

Reviewer's Responses to Questions

**Key Review Criteria Required for Acceptance?**

**Methods**

-Are the objectives of the study clearly articulated with a clear testable hypothesis stated?

-Is the study design appropriate to address the stated objectives?

-Is the population clearly described and appropriate for the hypothesis being tested?

-Is the sample size sufficient to ensure adequate power to address the hypothesis being tested?

-Were correct statistical analysis used to support conclusions?

-Are there concerns about ethical or regulatory requirements being met?

Reviewer #2: ok

Reviewer #3: 1. The response regarding the effective sample size of the trial is not sufficient. Since entomological sampling was conducted in only 50% of households, then the effective sample size for measurement of the endpoints is is 7 households x 14 clusters per arm. The additional 7 households per arm in which emanators were installed provide no information on the endpoints, therefore do not contribute to the sample size for the purpose of statistical power.

This needs to be addressed by

i. adjusting the sentence at line 265-7 to read "we assessed the impact of metofluthrin emanators on the entomological endpoints in approximately 100 houses per treatment arm, i.e. 50% of the households in which emanators were installed."

and ii. Re-running the Monte Carlo simulations to reflect the actual trial design, in terms of the information generated on intervention effect (i.e. 14 clusters, each with 7 houses) and updated all of the power estimates reported in the 'Power and sample size calculations' section accordingly.

2. Thank you for the additional information regarding the human landing catches. Please updated the sentence at line 304 accordingly: "Human landing counts (12) were conducted in 12 houses (4 per arm) selected based on high baseline entomological indices, resident willingness and ease of access".

**Results**

-Does the analysis presented match the analysis plan?

-Are the results clearly and completely presented?

-Are the figures (Tables, Images) of sufficient quality for clarity?

Reviewer #2: ok

Reviewer #3: Yes

**Conclusions**

-Are the conclusions supported by the data presented?

-Are the limitations of analysis clearly described?

-Do the authors discuss how these data can be helpful to advance our understanding of the topic under study?

-Is public health relevance addressed?

Reviewer #2: ok

Reviewer #3: Yes

**Editorial and Data Presentation Modifications?**

Reviewer #2: ok

Reviewer #3: (No Response)

**Summary and General Comments**

Reviewer #2: The revision is satisfactory.

Reviewer #3: The revised manuscript is much improved, and the authors have addressed the majority of my previous comments with the exception of two issues in the Methods section which still need to be addressed in a subsequent revision.

PLOS authors have the option to publish the peer review history of their article (what does this mean? ). If published, this will include your full peer review and any attached files.

**Do you want your identity to be public for this peer review?** For information about this choice, including consent withdrawal, please see our Privacy Policy .

Reviewer #2: No

Reviewer #3: No

**Figure resubmission:**
---

## [Editor Report · Decision Letter 2]

14 Aug 2025

Dear Prof. Manrique-Saide,

We are pleased to inform you that your manuscript 'Community deployment of metofluthrin emanators to control indoor Aedes aegypti: efficacy results from a crossover trial in Yucatan, Mexico.' has been provisionally accepted for publication in PLOS Neglected Tropical Diseases.

Best regards,

Ran Wang, M.D.

Academic Editor

Paul Mireji

Section Editor

Shaden Kamhawi

co-Editor-in-Chief

Paul Brindley

co-Editor-in-Chief

---

## [Editor Report · Acceptance letter]

Dear Prof. Manrique-Saide,

We are delighted to inform you that your manuscript, "Community deployment of metofluthrin emanators to control indoor Aedes aegypti: efficacy results from a crossover trial in Yucatan, Mexico.," has been formally accepted for publication in PLOS Neglected Tropical Diseases.

Best regards,

Shaden Kamhawi

co-Editor-in-Chief

Paul Brindley

co-Editor-in-Chief
